# Synthetic Data Generation & Multi-Step RL for Reasoning & Tool Use

**Anna Goldie & Azalia Mirhoseini**
Department of Computer Science
Stanford University
{agoldie,azalia}@cs.stanford.edu

**Hao Zhou & Irene Cai**
Google DeepMind
Mountain View, CA, USA
{haozhouml,irenecai}@google.com

**Christopher D. Manning**
Department of Computer Science
Stanford University
{manning}@cs.stanford.edu

## Abstract

Reinforcement learning has been shown to improve the performance of large language models. However, traditional approaches like RLHF or RLAIF treat the problem as single-step. As focus shifts toward more complex reasoning and agentic tasks, language models must take multiple steps of text generation, reasoning and environment interaction before generating a solution. We propose a synthetic data generation and RL methodology targeting multi-step optimization scenarios. This approach, called Step-Wise Reinforcement Learning (SWiRL), iteratively generates multi-step reasoning and tool use data, and then learns from that data. It employs a simple step-wise decomposition that breaks each multi-step trajectory into multiple sub-trajectories corresponding to each action by the original model. It then applies synthetic data filtering and RL optimization on these sub-trajectories. We evaluated SWiRL on a number of multi-step tool use, question answering, and mathematical reasoning tasks. Our experiments show that SWiRL outperforms baseline approaches by 21.5%, 12.3%, 14.8%, 11.1%, and 15.3% in relative accuracy on GSM8K, HotPotQA, CofCA, MuSiQue, and BeerQA, respectively. Excitingly, the approach exhibits generalization across tasks: for example, training only on HotPotQA (text question-answering) improves zero-shot performance on GSM8K (a math dataset) by a relative 16.9%.

## 1 Introduction

Large Language Models (LLMs) have demonstrated remarkable capabilities in Natural Language Processing (Gemini Team et al., 2024; Anthropic, 2024; OpenAI et al., 2024). However, they often struggle to answer complex queries that require reasoning and tool use across multiple steps (Wu et al., 2024), such as multi-hop question-answering, mathematical problem-solving, coding, and other agentic tasks, (Yang et al., 2018; Trivedi et al., 2022; Wu et al., 2024; Cobbe et al., 2021; Jimenez et al., 2024; Ehrlich et al., 2025; Li et al., 2022).

Traditional reinforcement learning (RL) approaches, such as RL From Human Feedback (RLHF) (Christiano et al., 2023), RL from AI Feedback (RLAIF) (Bai et al., 2022), and RL from Execution Feedback (RLEF) (Gehring et al., 2025), have focused on single-step optimization, leaving the challenge of multi-step tasks largely unaddressed. Many real-world problems require a sequence of interrelated actions; for example, when answering a challenging question, a model must determine not just what information to seek, but when to stop searching and synthesize its findings. Multi-step reasoning creates a compounding challenge, as incorrect intermediate steps often lead to incorrect final results, making it critical to maintain accuracy across the entire chain of actions or learn to effectively recover from such errors.

To address this challenge, we present Step-Wise Reinforcement Learning (SWiRL), an offline multi-step optimization technique. We consider a setting where the model has access to a tool, such as a search engine or calculator, and can run a sequence of tool use calls as needed to answer the question. Our goal is to teach the model how to decompose complex problems into a sequence of more manageable subtasks, when to call the tool, how to formulate a call to the tool, when to use the results of these queries to answer the question, and how to effectively synthesize its findings. In particular, we propose a two stage approach, in which we first generate multi-step synthetic data and then learn from these data using a step-wise reinforcement learning method. This approach has the key practical advantage that we can quickly generate large volumes of multi-step training data via parallel calls to avoid throttling the training process with slow tool use execution. In addition, this offline process enables greater reproducibility due to having a fixed dataset.

To generate multi-step synthetic training data, we provide an open-source LLM (Gemma 2 (Gemma Team et al., 2024b)) with access to a relevant tool (e.g., a search engine or calculator). We iteratively prompt the model to generate multi-step trajectories; at each step, the model is free to generate a chain of thought, and may either call a tool or produce a final answer, which we refer to as the model's action. If the model generates a tool use call, its query is automatically extracted from the overall response and executed in the environment, and the result is presented to the model in the next step. The trajectory ends when the model generates an answer to the original question, which it indicates using special markers. We convert each trajectory with $k$ actions into $k$ subtrajectories, containing the context from the beginning of the trajectory up to that action. We then use a step-wise reinforcement learning approach to optimize over this dataset, employing a generative reward model that evaluates each action in the context of its subtrajectory.

This granular approach enables us to apply direct feedback after each step of the trajectory, and to do so in a manner that is contextually aware. Unlike prior RL finetuning approaches used in frontier open-source models like DeepSeek-R1 (DeepSeek-AI and others, 2025) and Llama-3 (Grattafiori et al., 2024), we do not solely optimize for final performance, and use no golden labels; however, by optimizing for the reasonableness of each step given prior steps, SWiRL does in fact improve final performance.

In addition to evaluating SWiRL on challenging multi-hop question-answering and mathematical problem-solving tasks, we also study the generalization properties of this methodology. This is of key interest because there is an explosion of agentic applications for language models, and methods that generalize across datasets and tasks will be easier, cheaper and faster to adapt to new environments. We also measure the effectiveness of different synthetic data filtering strategies, study SWiRL's ability to generalize across datasets and tasks, measure the impact of model size and dataset size, and explore the mechanism driving these performance improvements.

Our contributions are as follows:

- We propose Step-Wise Reinforcement Learning (SWiRL), an approach to synthetic data generation and offline RL that advances multi-step reasoning and tool use.

- We demonstrate generalization across datasets. For example, training SWiRL on HotPotQA not only improves performance on the dataset itself, but also yields superior performance on other multi-hop question-answering datasets, e.g., 21.5% on GSM8K (Cobbe et al., 2021), 15.3% on BeerQA (Qi et al., 2021b), 11.1% on MuSiQue (Trivedi et al., 2022) and 14.8% on CofCA (Wu et al., 2024).

- We also show transfer across disparate tasks, namely mathematical reasoning to question-answering and vice versa. Training only on multi-hop HotPotQA question-answering improves performance on GSM8K (Cobbe et al., 2021) (a math dataset) by 16.9%, and training on GSM8K improves performance on HotPotQA (multi-hop question-answering) by 9.2%.

- We analyze the impact of synthetic data filtering strategies in a multi-step reasoning and tool use setting, and demonstrate that models learn best from datasets which have been filtered step-wise to ensure high-quality reasoning traces, but which are not filtered by outcome (correct final answer).

- We explore the impact of training dataset size and model size on SWiRL, observing that significant gains can be achieved even with just 1000 trajectories and that smaller models (Gemma-2-2b and 9b) can benefit from in-domain SWiRL, but do not display the same generalization as their larger counterpart, Gemma-2-27b.
- We demonstrate that SWiRL effectively improves the average process reward, even when evaluated on out-of-distribution tasks, suggesting that the downstream performance gains are driven by improved multi-step reasoning.

## 2 Methodology

Our methodology, Step-Wise Reinforcement Learning (SWiRL), consists of two stages. In the first stage, we generate and filter synthetic data. In the second stage, we use a step-wise reinforcement learning approach to optimize a generative base model on the synthetic trajectories. SWiRL does not require golden labels or human annotations, and instead relies entirely on model-based judgments for data generation, filtering, and RL optimization. The overall flow of our methodology is depicted in Figure 1 (Stage 1) and Figure 2 (Stage 2).

### 2.1 Multi-Step Data Collection

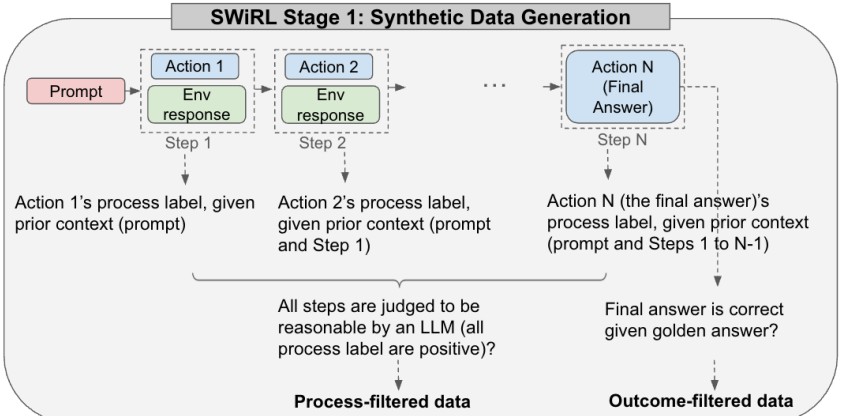

Figure 1: In SWiRL Stage 1, we generate and filter multi-step synthetic trajectories. At each step, the model is free to generate a chain of thought, call a tool such as a search engine or calculator, and/or produce an answer to original question. Process-filtered data corresponds to trajectories in which every step is judged to be reasonable by a model judge (Gemini 1.5 Pro Thinking). Outcome-filtered data corresponds to trajectories with a final answer that matches the golden label.

In Stage 1 (see Figure 1), we generate synthetic trajectories consisting of multiple steps of reasoning and tool use, which we use as training data for the step-wise RL methodology described in the next section. To compile a large-scale collection of synthetic trajectories, we augment a language model with a tool (e.g., a search engine or calculator), and iteratively prompt the model to generate multi-step trajectories. At each step, the model is asked to choose whether to call a tool or produce a final answer, and is always free to generate chains of thought (which it typically does). If the model generates a tool use call, it is parsed from the overall response, executed in the environment, and the result is presented to the model in the next step. See Appendix A for the prompt, which contains a question, explicit instructions regarding multi-step tool utilization, and the results of prior tool use calls.

For each multi-step synthetic trajectory, we define the following annotations. The trajectory itself is denoted by $\tau = (s_1, a_1, \ldots, s_K, a_K)$. The first state $s_1$ is the original prompt. Each following state $s_i$ contains the entire context so far, containing state $s_{i-1}$, action $a_{i-1}$, and the environment (tool call) response to $a_{i-1}$. Each action $a_i$ is the model response, given state $s_i$. The last action, $a_K$, is the model's answer to the original prompt.

In this work, we compiled a dataset of 50,000 synthetic trajectories seeded by 10,000 multi-step questions from the HotPotQA training set (Yang et al., 2018) (i.e., 5 trajectories per question), and a mathematical reasoning dataset of 37,500 synthetic trajectories seeded by the 7,500 questions in the GSM8K training set (Cobbe et al., 2021). Note that, for HotPotQA, we filtered out "Easy" questions, which can typically be answered with a single search query. To prevent synthetic trajectories from being excessively long, we set a maximum step count of 5 for HotPotQA questions, and 10 for GSM8K questions (which typically require 2-8 steps to solve).

Having compiled these datasets, we consider four different filtering strategies and measure their impact on performance (Figure 1): (1) No filtering; (2) Process filtering, where we retain trajectories in which each step was deemed reasonable given all previous steps. Concretely, a model (Gemini 1.5 Pro Thinking, in our case) is prompted to render a binary judgment as to whether action $a_i$ is reasonable given the context $s_i$. See Appendix A for our prompt. No golden labels are used; (3) Outcome filtering, where we select trajectories based solely on whether the final response, $a_K$, matches the golden answer; and (4) Process and outcome filtering, in which we take the intersection of both filtering approaches and retain only trajectories that exhibit both step-wise soundness and correct final outcomes.

Recent approaches to synthetic data distillation, such as Deep-Seek R1 (DeepSeek-AI and others, 2025), have demonstrated that synthetic data filtered for correct outcomes can lead to good performance with single-step RL and supervised finetuning (SFT). In this work, we sought to explore whether this pattern would hold in a multi-step, tool use setting, and to explore the impact of both outcome and process filters. Like these prior work, we observed that filtering multi-step trajectories for correctness was effective for SFT, and in fact critical for good performance. However, we found that SWiRL, unlike SFT, can learn even from trajectories that end in incorrect final answers. In fact, we achieve our best results by including process-filtered data, regardless of the correctness of the outcome.

## 2.2 Step-Wise Reinforcement Learning Methodology

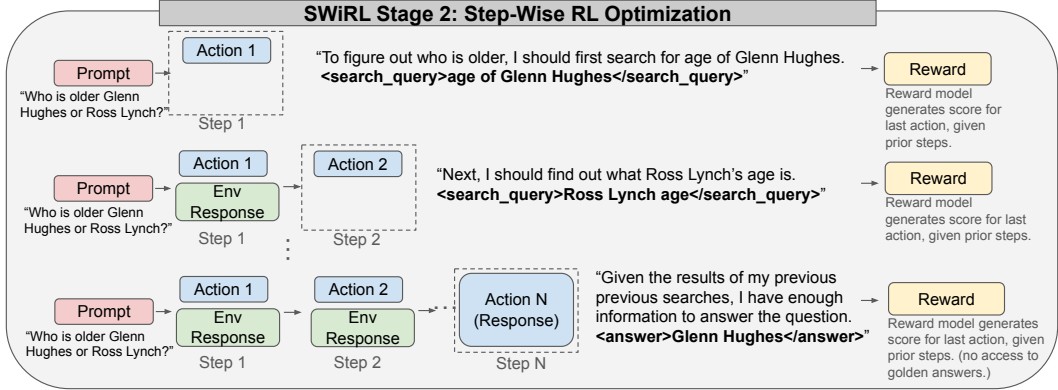

Figure 2: In SWiRL Stage 2, we perform step-wise RL to train on the synthetic multi-step trajectories from Stage 1. Each step contains an action, which corresponds to a tool call or the final response. The model is free to generate chains of thought during each step. The environment responses are captured in the prior steps of the synthetic trajectories, which were generated offline. Granular feedback is provided by a generative reward model, which is used to perform RL optimization directly on each action, given the prior context.

As shown in Figure 2, we propose a RL approach capable of learning effectively from the synthetic multi-step trajectories generated in Stage 1. At each step, a base model is optimized to predict either the next intermediate step or the final response based on preceding context. At each step $i$, the model has access to the full contextual history, including the original prompt, all previous model-generated steps and any applicable environment response corresponding to those steps.

Thus, our objective function is the expected sum of stepwise rewards:

$$J(\theta) = E_{s \sim \mathrm{T}, \, a \sim \pi_\theta(s)} \left[ R(a|s) \right]$$

Here, $\pi_\theta$ is the base model parametrized by $\theta$, which is finetuned via SWiRL (Note that we also use $\pi_\theta$ to generate synthetic data.) T denotes the set of all states in the synthetic multi-step trajectories, i.e. each incremental state $s$ within each trajectory $\tau$. The reward signal $R(a|s)$ is derived from a generative reward model, specifically Gemini 1.5 Pro in our experiments, which assesses the quality of the generated response $a$ given the context $s$. No golden labels are used.

We optimize the above expected reward using the same policy gradient algorithm as used in Gemma 2 for optimizing the human feedback reward (Gemma Team et al., 2024a;b). Our granular, step-by-step finetuning paradigm enables the model to learn both local decision-making (next-step prediction) and global trajectory optimization (final response generation) while being guided by immediate feedback on the soundness of each prediction.

## 2.3 Step-Wise Inference-time Evaluation

As shown in Figure 3, at inference time, we iteratively prompt the model to either call a tool or produce a final answer. If the model generates a search query (indicated by <search_query> </search_query> tags), we parse out that query, embed it with a Gecko model, perform a nearest neighbor lookup in the corresponding vector database, and inject the retrieved article into the model's context window. If the model generates a calculator tool call (indicated by <math_exp> </math_exp> tags), we parse out the mathematical expression, execute it with a SymPy interpreter, and inject the calculated results into the context window. This process terminates when the model either produces an answer (signaled by producing <answer> </answer> tags) or reaches the maximum number of queries (5 for question-answering datasets, and 10 for mathematical reasoning datasets). See Appendix E for example trajectories.

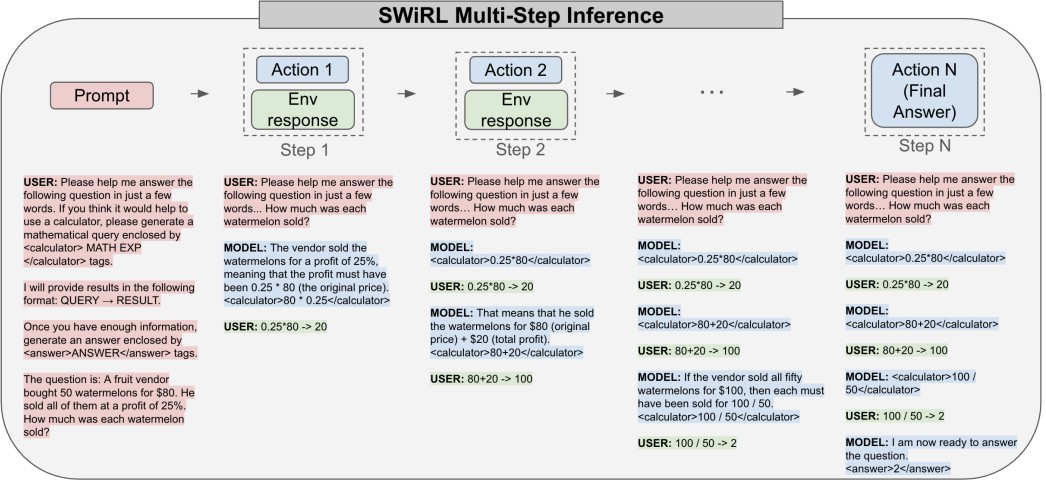

Figure 3: SWiRL Multi-Step Inference. At inference time, we iteratively prompt the model to call available tools as many times as necessary (up to a limit) before answering the original question. Here, prompts are truncated for clarity and to adhere to space limitations, but full trajectories are available in Appendix E.

# 3 Related Work

**Reinforcement Learning for LLM Finetuning.** One prominent approach, Reinforcement Learning from Human Feedback (RLHF) (Ouyang et al., 2022; Christiano et al., 2023), con-

sists of training a reward model on human preference labels at the response level, followed by RL optimization using Proximal Policy Optimization (PPO) (Schulman et al., 2017). Building upon this framework, Reinforcement Learning with AI Feedback (RLAIF) (Bai et al., 2022) has emerged as a scalable alternative that leverages AI models to generate feedback based on predefined principles or constitutions, reducing the need for costly human annotations. RL from Execution Feedback (RLEF) (Gehring et al., 2025) uses environment feedback, such as pass rate on coding test cases, to calculate the reward, which it then optimized via PPO. Besides PPO, other RL optimizations, such as Direct Preference Optimization (DPO) (Rafailov et al., 2023) and its successors (e.g., Azar et al. (2023); Ethayarajh et al. (2024); Meng et al. (2024); Lanchantin et al. (2025)) as well as GRPO (Shao et al., 2024) have also proven to be effective for finetuning LLMs to maximize a target reward. A limitation of the above approaches is that they focus on single-step optimization with the reward being calculated only at the end of the episode, leading to suboptimal performance for multi-step optimization (Liu et al., 2024; Wang et al., 2024). In SWiRL, we focus on scenarios where multiple steps of reasoning and tool calls are necessary prior to generating a response. Unlike the above methods, SWiRL enables the model to receive feedback on its granular stepwise actions which leads to better multi-step reasoning and tool use across longer horizons.

**Multi-step Optimization with RL.** Recent work including DQO (Liu et al., 2024) and OREO (Wang et al., 2024) propose offline reinforcement learning to improve multi-step reasoning for LLMs. However, neither focuses on enhancing a model's ability to use tools or interact with an external environment. Additionally, unlike our approach, which optimizes at the (reasoning) step level, DQO relies on token-level actions, which as shown in (Wang et al., 2024), are generally less effective than step-level actions. Moreover, OREO requires training a separate value network and policy, and relies on iterative co-optimization of both models. The process of maintaining, training, and serving these two models can be prohibitively expensive, particularly for larger models. PRIME (Cui et al., 2025) proposes an online approach to improve multi-step reasoning, but does not enable tool use or offline training. Tulu-3 (Lambert et al., 2025) uses verifiable rewards to train a language model to do better at math, but requires access to golden labels.

**Reasoning Improvement with Synthetic Data**. Several approaches have been proposed for generating synthetic reasoning data. These methods either rely on golden labels to filter the data or use a combination of golden labels and process or outcome reward models (Zelikman et al., 2022; Singh et al., 2024). For example, STaR (Zelikman et al., 2022) generates chain-of-thoughts (CoT) for reasoning questions, filters for those that result in correct answers, and performs Supervised Fine-Tuning (SFT) on those reasoning traces. The paper also proposes an augmentation technique called "rationalization", in which for each question the model answered incorrectly, the model is provided with the correct answer and prompted to generate a CoT that leads to that answer. Rejection finetuning (RFT) (Yuan et al., 2023) is another method that relies on collecting reasoning traces from the model and using those with correct outcomes for SFT. ReST (Gulcehre et al., 2023) demonstrates strong performance on machine translation by iteratively generating data and then finetuning on that data using either a supervised or reinforcement learning objective. $ReST^{EM}$ (Singh et al., 2024) is an extension of ReST which outperforms training on human data alone for math and coding evaluations, but which plateaus after a few iterations, presumably due to overfitting. Our method also uses a model-based approach to generate multi-step trajectories. However, we show that using a model to label the steps within each reasoning trajectory leads to higher out-of-domain generalization than using only the trajectories which contain correct final answers, meaning that we do not require golden labels. In addition, we enable the model to use tools iteratively to perform multi-hop question answering and mathematical reasoning.

**Process vs. Outcome Based Optimization**. There have been a number of attempts to compare the effectiveness of process and outcome-based approaches in the domain of math and reasoning (Lightman et al., 2023; Uesato et al., 2022; Snell et al., 2024). For example, (Lightman et al., 2023) showed that (Outcome Reward Models) ORMs are more effective than (Process Reward Models) PRMs at the task of ranking samples from a fixed generator model, whereas Uesato et al. (2022) demonstrated that outcome supervision yields comparable accuracy to process supervision at lower cost, but that the reasoning traces from the resulting model exhibit lower fidelity. Both rely on expensive human annotations and golden labels,

and do not explore the impact of PRMs and ORMs in reinforcement learning optimization, or the differential effect of data filtering on supervised vs. RL optimization objectives.

## 4 Experiments

| Datasets
Metrics | HotpotQA
PM† | CofCA (Avg)
PM† | MuSiQue
PM† |
|---|---|---|---|
| **Proprietary LLMs** | | | |
| **GPT-4** | 74.8 | 51.9 | 63.9 |
| **GPT-3.5** | 62.8 | 40.7 | 53.1 |
| **Gemini 1.0 Pro** | 63.5 | 33.3 | 46.9 |
| **Bing Chat** | 72.1 | 41.6 | 52.3 |
| **O1-preview** | **76.9** | **58.5** | **67.9** |
| **Open Source LLMs** | | | |
| **Llama 2-7b** | 38.5 | 28.9 | 34.2 |
| **Mistral-7b** | 34.9 | 25.6 | 29.2 |
| **Qwen 2-7b** | 39.3 | 30.7 | 33.5 |
| **Base Gemma 2-27b** | 58.6 | 31.7 | 35.4 |
| **SWiRL Gemma 2-27b (Ours)** | **67.8** | **39.3** | **43.6** |

Table 1: Comparison of Accuracy (**PM†**: Partial Match) across Multiple Datasets: **HotpotQA**, **CofCA** (Average of 2-hop, 3-hop, and 4-hop), and **MuSiQue**. Baseline results were drawn from Wu et al. (2024). The Gemma-2 models, both SWiRL and the base model, were not given access to the context documents, but were allowed to sequentially query a vector database. The SWiRL model was trained on HotPotQA using process-filtered data, and for consistency with baseline results, evaluated on GPT-4o with the same prompts as Wu et al. (2024) on 300 randomly subsampled questions. See Appendix F for example ids.

### 4.1 Evaluation Datasets

To evaluate performance on multi-step search tool use, we selected five challenging multi-hop question-answering and mathematical reasoning datasets:

- **HotPotQA** (Yang et al., 2018) is comprised of multi-hop questions from a variety of domains. Human annotators constructed the questions to be answerable only by combining information from multiple paragraphs of Wikipedia.

- **MuSiQue** (Trivedi et al., 2022) is a multi-hop question-answering dataset constructed by chaining together multiple single-hop questions.

- **CofCA** (Wu et al., 2024) is a multi-hop dataset constructed to be answerable only by querying a counterfactual version of Wikipedia. It contains 2- to 4-hop questions.

- **BeerQA** (Qi et al., 2021a) is an extension of HotPotQA designed to include an even greater number of hops than the original dataset.

- **GSM8K** (Cobbe et al., 2021) is a dataset composed of grade school math word problems, which typically take 2-8 steps to solve.

For question-answering datasets, we set up a vector database containing all articles from each data split using Gecko-1B with 768-dimensional embeddings (English) (Lee et al., 2024).

For the experiments in Table 1, we follow the same procedure as Wu et al. (2024), evaluating performance on 300 randomly subsampled examples from the target dataset, using the same language model as a judge (GPT4o) and the same prompt. For every other experiment in this paper, we used Gemma-2-27b as our judge, as this was more cost effective, with the exception of GSM8K for which we used Gemini 1.5 Pro as it exhibited noticeably better numeric evaluation. Model-based evaluation is emerging as a scalable and less brittle alternative to exact match and F1 metrics (Zheng et al., 2023; Gu et al., 2025), but does

introduce a new source of stochasticity into the evaluation. See Appendix D for our own manual inspection and error analysis of three different model judges.

As described in Section 2.3, for each question, we iteratively prompt the model to either call a tool or produce a final answer, and limit the maximum number of queries to 5 for question-answering datasets, and 10 for mathematical reasoning datasets.

## 4.2 Results and Discussion

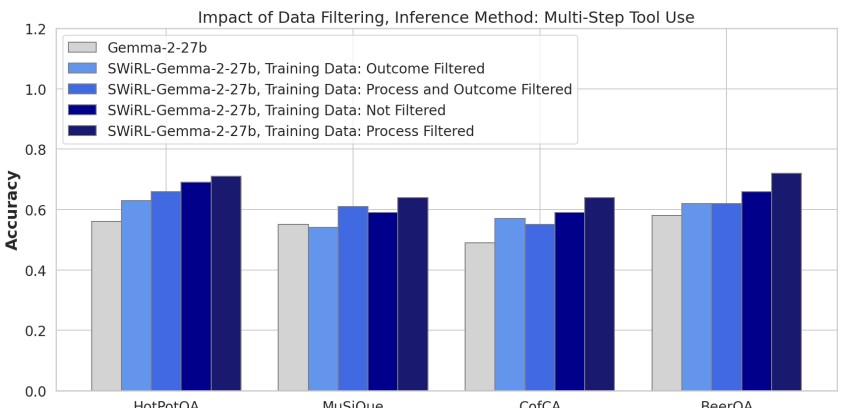

Figure 4: Impact of Data Filtering on Model Performance. Synthetic data for training is derived from HotPotQA. SWiRL learns to perform multi-hop question answering even when trained on unfiltered synthetic data. SWiRL's best performance comes from training on process-only filtered data, where the data is selected based on the soundness of each step within its reasoning traces, but which includes both correct and incorrect responses.

**Impact of Data Filtering on Model Performance:** We evaluated the influence of various filtering mechanisms on downstream task accuracy, as shown in Figure 4. Concretely, we consider 4 different types of filtering: no filtering, outcome-based filtering that ensures correct final answers, process-based filtering that ensure that each step is correct as judged by a model, and both process and outcome-based filtering.

In all experiments, we fix the number of trajectories used for finetuning (with the exception of our ablation study on the impact of scaling dataset size), and we provided all models with access to an appropriate tool. Notably, process-only filtering consistently yields the highest accuracy, suggesting that focusing on the procedural aspects of data refinement is more important than the correctness of a training trajectory. While both unfiltered and filtered data demonstrated an improvement over the baseline model, filtering for correctness usually harms performance; with the exception of MuSiQue, outcome-filtered or outcome and process-filtered data is less effective than unfiltered data. We hypothesize that this is because SWiRL actually benefits from having access to both positive and negative examples. These results underscore the relative unimportance of outcome-based filtering, which requires golden labels. They also demonstrate that our process RL method can effectively learn from even trajectories with incorrect final answers.

**Generalization Across Disparate Tasks:** To measure generalization across training tasks, we evaluated the mathematical reasoning capabilities of a model trained on multi-hop question-answering with search tool use (HotPotQA). Specifically, we evaluated the performance of this model on GSM8K, a mathematical reasoning task, providing the model with a SymPy interpreter to use as a calculator. This experiment was run on a different random subsample of 300 examples. As shown in Table 2, applying SWiRL on out-of-distribution data and tasks still improves performance.

**Effect of Tool Use:** As discussed in Section 2.3, at inference time, we use the proposed multi-step eval as shown in Figure 3 and we iteratively prompt the model to make tool calls as necessary to answer the question. As shown in Figure 5, both base and SWiRL models

|  | GSM8K (math) | HotPotQA (qa) | CofCA (qa) | BeerQA (qa) | MuSiQue (qa) |
|---|---|---|---|---|---|
| Base Model | 0.65 | 0.65 | 0.54 | 0.59 | 0.45 |
| SWIRL on GSM8K (math) | 0.79 | 0.71 | 0.56 | 0.68 | 0.49 |
| SWIRL on HotPotQA (qa) | 0.76 | 0.73 | 0.62 | 0.68 | 0.50 |

Table 2: SWiRL Generalization Performance. Finetuning on synthetic traces from HotPotQA or GSM8K improves performance on both in-distribution and out-of-distribution tasks. Interestingly, training on a different domain and tool (e.g. math and a calculator) improves performance on question-answering with a search engine and vice versa, suggesting the effectiveness of SWiRL in improving general multi-step reasoning and tool use capability.

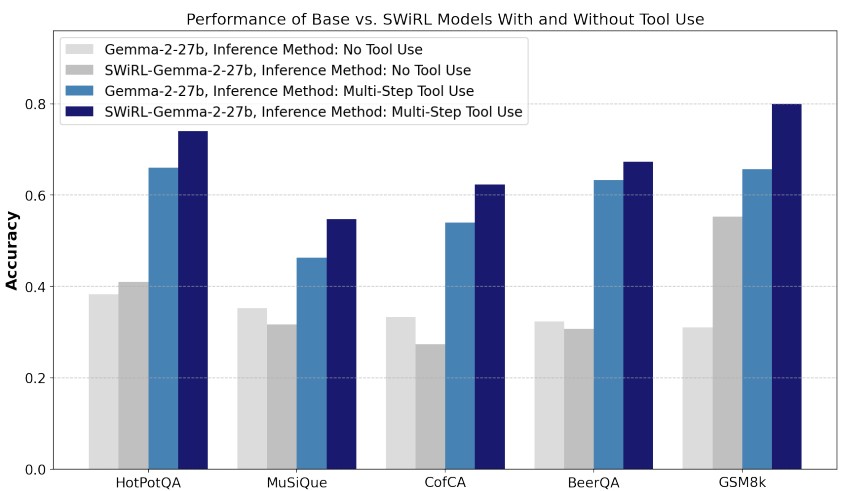

Figure 5: Performance of SWiRL With and Without Multi-Step Tool Use. SWiRL's multi-step tool use inference improves the performance of both the base model and the SWiRL-finetuned model, but benefits the latter substantially more. Even without access to a tool, the SWiRL model outperforms, suggesting that SWiRL training improves the model's ability to break down complex problems into multiple manageable subtasks.

improve with SWiRL's multi-step tool use inference, but SWiRL-training offers even further improvements. Notably, the SWiRL model exhibits substantial improvements, even without access to a tool, suggesting that SWiRL training improves the model's ability to break down complex problems into multiple manageable subtasks.

**Impact of Scaling Finetuning Dataset and Model Size:** Our experiments on scaling the fine-tuning dataset size reveal a clear trend: SWiRL has the ability to leverage larger datasets, even when using only process-filtered data, as shown in Figure 6. As the fine-tuning dataset size increases, a consistent enhancement in model performance is observed across our target multi-step reasoning tasks. While a limited dataset of 100 data points appears insufficient for the model to effectively generalize, a significant improvement is evident with 1,000 data points, showing solid gains across all datasets. Furthermore, scaling up to 10,000 data points continues to yield further performance enhancements, confirming the efficacy of our method in capitalizing on larger datasets for improved reasoning capabilities.

We also varied model size, observing that smaller models (2b and 9b) may benefit from in-domain SWiRL, but do not display the same generalization as their larger counterpart, Gemma 2 27b. See results in Appendix B.

**Effect on Mean Process Label Accuracy**: In the previous subsections, we evaluated the effect of SWiRL on downstream task accuracy. Here, we take a deeper look to understand how SWiRL achieves these performance improvements. In Table 3, we show the average process

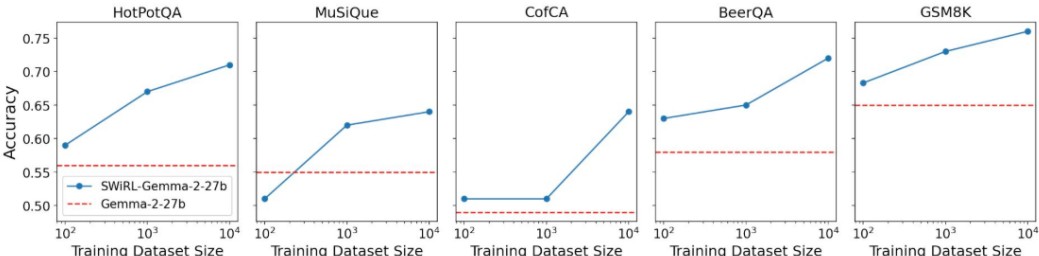

Figure 6: Performance as a Function of Synthetic Dataset Size. Synthetic training data is derived from HotPotQA, and accuracy is evaluated by Gemma 2 27b. As we scale the dataset size, we observe consistent improvements in model performance. With only 1000 data points, the model robustly improves both on in- and out- of distribution datasets.

label accuracy for the baseline model vs. a SWiRL finetuned model on 500 trajectories (seeded by 100 questions) for both HotPotQA and GSM8K. To calculate the score per step, we use the same model and prompt as we used for process filtering, as described in Section 4. We take a macro-average of the process label scores within and then across trajectories. We observe that both for in-distribution and out-of-distribution tasks, the SWiRL model generates trajectories with higher average process labels, suggesting that the higher final accuracies are driven by better multi-step reasoning.

|  | HotPotQA (in distribution) | GSM8K (out of distribution) |
|---|---|---|
| Base (Mean Process Label) | 82.5% | 87.5% |
| SWiRL on HotPotQA (Mean Process Label) | 91.0% | 91.6% |

Table 3: Impact of SWiRL on Process Correctness. After our multi-step RL optimization, we observe that the average correctness of each step improves over the base model on both in- and out- of distribution tasks.

## 5 Conclusion

In this work, we propose a synthetic data generation and offline reinforcement learning approach to multi-step reasoning and tool use. This approach outperforms baselines by an average 15% across challenging multi-hop question-answering and mathematical reasoning tasks. We explore the effect of different data filtering strategies in a multi-step, tool use setting, and find that our RL approach is effective even on unfiltered data, but performs best on process-filtered data. Unlike supervised finetuning, our RL approach can learn from trajectories with incorrect final answers and actually benefits from the presence of a mixture of both correct and incorrect final answers. SWiRL demonstrates strong generalization properties, improving performance on mathematical reasoning (GSM8K) by 16.9% when trained on multi-hop question-answering (HotPotQA) and 9.2% vice versa.

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

# A  Prompts for Synthetic Data Generation, Filtering, and Evaluation

In this work, we use the following prompts for data generation, filtering, and evaluation.

| Prompt Type | Prompt Text |
|---|---|
| **Prompt for Multi-Step Synthetic Data Generation for Question-Answering with Search Tool Use** | `<start_of_turn>`user
Please help me answer the following question in just a few words. If you think it would help to do a search, please generate a search query enclosed by `<search_query>` QUERY `</search_query>` tags.
Some questions may require multiple searches in order to answer, so I will allow you to make up to {} sequential queries before answering the question.
Please do not repeat queries you have already issued, as this is a waste of time.
I will provide search results in the following format:
`QUERY → RESULT`.
Once you have enough information, generate an answer enclosed by `<answer>`ANSWER`</answer>` tags.
Please either issue a search query or answer the question, but not both.
The question is: {}
`<end_of_turn>` |

| Prompt Type | Prompt Text |
|---|---|
| **Prompt for Multi-Step Synthetic Data Generation for Mathematical Reasoning with Calculator Tool Use** | `<start_of_turn>`user
Please help me answer the following question in just a few words. If you think it would help to use a calculator, please generate a mathematical query enclosed by `<math_exp>` MATH EXP `</math_exp>` tags.
Some questions may benefit from using a calculator multiple times in order to answer, so I will allow you to make up to {} sequential queries before answering the question.
Please do not repeat queries you have already issued, as this is a waste of time.
I will provide results in the following format:
`QUERY → RESULT`.
Once you have enough information, generate an answer enclosed by `<answer>`ANSWER`</answer>` tags.
Please either issue a search query or answer the question, but not both.
The question is: {}
`<end_of_turn>` |

| Prompt Type | Prompt Text |
| --- | --- |
| **Prompt for Process-Filtering on Multi-Step Search Tool Use Trajectories** | `<start_of_turn>`user
My boss asked me to answer the following question with the help of a search engine: {}
This means that I might need to decompose the question into a sequence of searches before being able to answer the question.
I am trying to learn how to do this more effectively, so please provide feedback on my last message.
Please take a look at our conversation so far: {}
When evaluating a message, please only consider the last message and do not penalize or reward me for previous messages.
When evaluating an answer, please consider only whether the answer follows from the search results, and not whether you believe the answer to be correct.
If there is not enough information from the search results to answer the question, you should rate any answer as "BAD". Pay close attention as it may initially seem like the answer is present when it is not.
When evaluating a search query, please consider whether it is likely to help me answer the original question.
Explain your reasoning and then answer with either "GOOD" or "BAD".
`<end_of_turn>` |

| Prompt Type | Prompt Text |
| --- | --- |
| **Prompt for Evaluation / Outcome-Filtering on Multi-Step Trajectories with Search Tool Use** | `<start_of_turn>`user
I need you to help me grade the answer to the following question: "{}".
The answer key says: {}, and my answer is {}. Am I correct?
Please explain your reasoning and then answer "YES" or "NO".
Do not use your own knowledge to the decide, but simply check whether I gave the answer in the answer key.
`<end_of_turn>` |

| Prompt Type | Prompt Text |
| --- | --- |
| **Prompt for Evaluation / Outcome-Filtering on Multi-Step Trajectories with Calculator Tool Use** | `<start_of_turn>`user
I need you to help me grade the answer to the following question: "{}".
The answer key says: {}, and my answer is {}. Am I correct?
Please explain your reasoning and then answer "YES" or "NO".
There are multiple ways to write the same answer. For example, "10", "10.00", "$10", and "$10.00" are all equivalent.
`<end_of_turn>` |

# B   Impact of Model Size on Effectiveness of SWiRL

The trend is that models are growing in parameter count over time (Sevilla et al., 2022), so measuring the impact of model size on the effectiveness of a method can provide insight into its longevity and future impact. It is also interesting to see whether larger models are able to learn more general patterns from the training process, and therefore exhibit greater transfer learning across datasets and even domains (e.g. math vs. question-answering). As shown in Figure 7, SWiRL demonstrates a clear performance boost over the baseline Gemma 2-27b model, showcasing consistent improvements across both in-domain (HotPotQA) and out-of-domain datasets (MuSiQue, COFCA, and BeerQA); while the 2b and 9b Gemma models also exhibit enhanced performance on in-domain data, their generalization performance on out-of-domain data is less consistent. This suggests that the effectiveness of SWiRL grows with increased model size, which is consistent with the observation that methods such as RLHF (Ouyang et al., 2022) and RLAIF (Bai et al., 2022) are more effective for larger models.

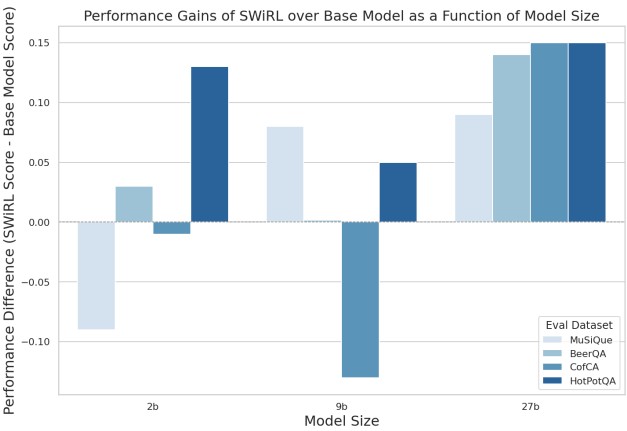

Figure 7: SWiRL Performance vs. Model Size. Synthetic data for training is derived from HotPotQA. Step-Wise RL finetuning robustly improves performance over baseline for the 27b model across both in-domain (HotPotQA) and out-of-domain datasets (MuSiQue, CofCA, and BeerQA). However, while the in-domain improvements hold for smaller models, the out-of-domain performance is mixed, suggesting that the relative effectiveness of SWiRL is higher for larger models.

## C Comparison of Supervised Finetuning and SWiRL

Figure 8 compares the performance of Supervised Fine-Tuning (SFT) and SWiRL on down-stream tasks. The results show that SFT leads to worse overall performance when compared to SWiRL across all data filtering strategies. We observe that SFT performs better if we apply it to data that is both process and outcome-filtered, rather only process-filtered. However, interestingly, SWiRL learns best from data that is only process-filtered. We attribute this to SFT's tendency to memorize, rather than generalize (Chu et al., 2025; Setlur et al., 2024), which can hinder the model's performance on new, unseen scenarios. In contrast, SWiRL has the ability to improve model performance by targeting per-step reward maximization. SWiRL enables the model to develop a deeper understanding of the necessary steps of query generation and retrieval, which leads to enhanced planning and generalization.

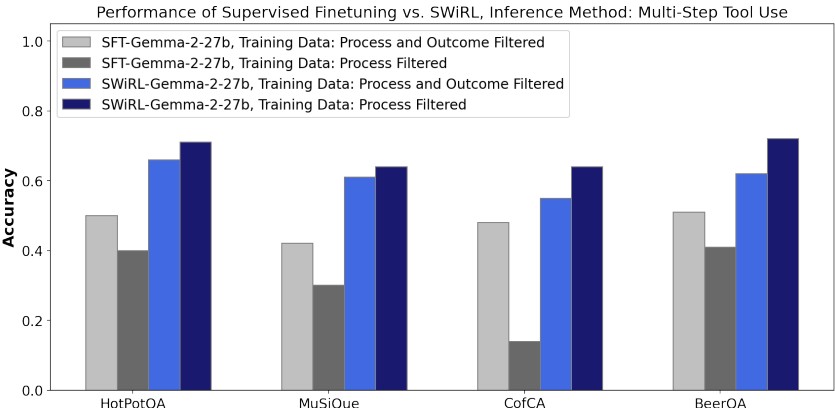

Figure 8: Comparison of SFT and SWiRL. Synthetic data for training is derived from HotPotQA, and to derive accuracy, Gemma 2 27b evaluates whether the model's answer matches the golden answer. SWiRL greatly benefits from process-only filtered traces, and unlike SFT, is capable of learning from traces with both correct and incorrect outcomes.

## D  Error Analysis of Three LLM Judges

Table 4: Error Rates for Gemma-2-27b Judgments on HotPotQA (N=100)

| Metric | Rate (%) |
|---|---|
| False Positive Rate (FPR) | 4 |
| False Negative Rate (FNR) | 1 |

Table 5: Manual Analysis of LLM Math Grading Accuracy (N=100)

| Model | FPR | FNR | Notes |
|---|---|---|---|
| Gemma-2-27b | 15 | 0 | Overly permissive ("nice"); all errors involved units. |
| GPT-4o | 0 | 10 | Overly harsh; all errors involved units. |
| Gemini 1.5 Pro | 4 | 0 | Accurate, slightly permissive; all errors involved units. |

To evaluate the suitability of language models to serve as evaluators (i.e., check the correctness of a model answer, given a golden answer), we manually checked the correctness of 100 model judgments from Gemma-2-27b on HotPotQA questions. As shown in Table 4, we found that the error was relatively low (4% false positives and 1% false negatives), justifying the use of this low cost open-source model as our LLM judge.

However, we noticed that Gemma-2-27b made more errors when it came to numeric quantities, so we decided to run a separate analysis for GSM8K, manually evaluating 100 model judgments each for three language models (Gemma-2-27b, GPT-4o, and Gemini 1.5 Pro). Interestingly, we found that Gemma-2-27b tended to be overly "permissive" in its grading, but had zero false negatives, whereas GPT-4o had a relatively high false negative rate but no false positives. We also observed that relative results were consistent across model judges; if GPT-4o gave a higher accuracy score to a particular model, Gemma-2-27b did as well, even if the absolute scores differed. To reduce noise, we chose to use Gemini 1.5 Pro as the LLM judge for GSM8K, in spite of its higher cost.

# E  Example Trajectories

For clarity, we provide example trajectories from question-answering and mathematical reasoning datasets, which showcase both search and calculator tool use.

| Trajectory Type | Trajectory Text |
|---|---|
| **Example Multi-Hop Question-Answering Trajectory with Search Tool Use Generated by SWiRL model on HotPotQA** | \<start_of_turn\>user
Please help me answer the following question in just a few words. If you think it would help to do a search, please generate a search query enclosed by \<search_query\> QUERY \</search_query\> tags.
Some questions may require multiple searches in order to answer, so I will allow you to make up to 5 sequential queries before answering the question.
Please do not repeat queries you have already issued, as this is a waste of time.
I will provide search results in the following format:
QUERY → RESULT.
Once you have enough information, generate an answer enclosed by \<answer\>ANSWER\</answer\> tags.
Please either issue a search query or answer the question, but not both.
The question is: What company published both The Scorch Trials and The Death Cure?
\<end_of_turn\>
\<start_of_turn\>model
\<search_query\>the scorch trials publisher
\</search_query\>
\<end_of_turn\>
\<start_of_turn\>user
the scorch trials publisher -¿ The Scorch Trials is a 2010 young adult post-apocalyptic dystopian science fiction novel written by American author James Dashner and the second book, fourth chronologically, in "The Maze Runner" series. The novel was published on September 18, 2010 by Delacorte Press. It is preceded by "The Maze Runner", and followed by "The Death Cure". A was released on September 18, 2015 by 20th Century Fox.
\<end_of_turn\>
\<start_of_turn\>model
\<search_query\>The Death cure publisher
\</search_query\>
\<end_of_turn\>
\<start_of_turn\>user
The Death cure publisher -¿ The Death Cure is a 2011 young adult dystopian science fiction novel written by American writer James Dashner and the third book, fifth chronologically, in the "Maze Runner" series. It was published on October 11, 2011 by Delacorte Press and was preceded by "The Maze Runner" and "The Scorch Trials" and followed by the series prequels, "The Kill Order and The Fever Code." \<end_of_turn\>
\<start_of_turn\>model
¡answer¿Delacorte Press¡/answer¿ \<end_of_turn\>\<eos\> |

| Trajectory Type | Trajectory Text |
|---|---|
| **Example Multi-Step Math Reasoning Trajectory with Calculator Tool Use Generated by SWiRL model on GSM8K** | `<start_of_turn>`user
Please help me answer the following question in just a few words. If you think it would help to use a calculator, please generate a mathematical query enclosed by `<math_exp>` QUERY `</math_exp>` tags.
Some questions may benefit from using a calculator multiple times in order to answer, so I will allow you to make up to 10 sequential calls before answering the question.
Please do not repeat queries you have already issued, as this is a waste of time.
I will provide search results in the following format:
`QUERY → RESULT`.
Once you have enough information, generate an answer enclosed by `<answer>`ANSWER`</answer>` tags.
Please either issue a math query or answer the question, but not both.
The question is: Natalia sold clips to 48 of her friends in April, and then she sold half as many clips in May. How many clips did Natalia sell altogether in April and May?
`<end_of_turn>`
`<start_of_turn>`model
`<math_exp>`48 / 2 `</math_exp>`
`<end_of_turn>`
`<start_of_turn>`user
48 / 2 -¿ 24.0
`<end_of_turn>`
`<start_of_turn>`model
`<math_exp>`48 + 24`</math_exp>`
`<end_of_turn>`
`<start_of_turn>`user
48 + 24 -¿ 72.0 `<end_of_turn>`
`<start_of_turn>`model
¡answer¿72¡/answer¿ `<end_of_turn><eos>` |

# F  Example Ids

For reproducibility, we provide the ids of the examples used for evaluation.



**HotPotQA Example Ids**

5ab46a295542990594ba9c1c, 5a85ad925542997b5ce40023, 5a8353285542996488c2e40d, 5a79be835542994bb9457052,
5a82a0fc55429954d2e2eb87, 5adc2c9355429947ff173904, 5ab1e71b554299340b525421, 5a7790ac5542992a6e59def9,
5a83e4195542990548d0b243, 5ab3239b554299194fa93574, 5ae1a460554299234fd042a8, 5a81e075554299676cceb128,
5a7f714c5542992097ad2f6e, 5ab639c055429953192ad2aa, 5a7c1fe4554299683c1c62cf, 5ab7cff355429928e1fe391e,
5aba5b2455429939ce03dc9c, 5a7173b45542994082a3e83c, 5a90049d55429933b8a20468, 5a8e0d7e55429995085b373b4,
5adfbf3155429906c02daa29, 5abf1fed5542990832d3a127, 5addf6415542990dbb2f7f25, 5a8138c155429938b6142300,
5a7ae77b554299042af8f6b0, 5ae293fb5542996483e649fe, 5ae40a8b55429970de88d8a9, 5ab457445542991751b4d748,
5a77d6025542995d83181301, 5a89c2715542993b751ca990, 5a7a4d845542990783324f04, 5ae0ef5e5542990adbacf6df,
5a72321f55429971e9dc934a, 5ac440355542995c82c4ad0d, 5a7dd8625542990b8f503ae8, 5ab48dd555429917779162cd9,
5abc3948554299700f9d782b, 5a8a7cb255429930ff3c0df8, 5ae1178e5542997b2ef7d0d6, 5abd08ae554299700f9d7980,
5ab1e5975542997061209590, 5a74dca85542996c70cfae1f, 5ab8348d55429934fafe6d13, 5a78f1ef554299747737f7919,
5ac16eb355429964131be1f5, 5ae5e12d55429929b08079e4, 5ade6bbf5542997c77adee24, 5adf573c5542995534e8c798,
5a8901d9554299515336125b, 5a89fd9e55429970aeb701e8, 5a7917d9554299747737f7982, 5adc1017554299438c868d20,
5a8da5c355429941ae14dffe, 5a8cad265542996e8ac88b19, 5add4ae25542992200553a88, 5ae026eb55429924de1b703a,
5a74fcbe5542996c70cfae67, 5adfa8ac55429942ec259add, 5adbf4555542994650320c18, 5ac31609554299741d48a1c0,
5a7b65bf55429931da12ca86, 5a73870455429905862fe051, 5a8b009755429950cd6afc40, 5ae62b2d5542992ae0d1625b,
5a7b5d795542992d025e6825, 5ab3185755429976abd1bc5f, 5ac0464755429996f0d89cb70, 5a89138255429951533612af,
5a85d69f5542997175ce2062, 5a82dfa455429940e5e1a938, 5a8730355542991e7718170f, 5a85b3455542994c784ddb4d,
5a8658c4554299211dda2b02, 5abd9fa55542996e802b4809, 5ab268aa5542993be8fa9908, 5ae5dcc755429929b08079d8,
5a727ef15542992359bc30c5, 5a8e2ba85542995a26add474, 5a84f9465542991dd0999e36, 5a87099455429960ec39b704,
5a864d835542994775f6073c, 5ab9bf3b554299743d22ebe6, 5a864dfc5542994775f6073f, 5a871ce055429960ec39b749,
5a8bd3375542997f31a41dd3, 5ab277965542993be8fa9919, 5abcea8355429114383a194, 5a897561554299515336130b,
5adfdf4a55429906c02daa7c, 5ae265bb5542992decbdccea, 5a84b3035542994a431d1a91, 5a77280b5542994aec3b71ff,
5ae4d413554299086326488, 5a76de035542994aec3b718d, 5a7d2045554299452d57bb09, 5abc7af15542993a06baf8ed,
5abddeb55542991f66106083, 5a8218855542990a1d231f4e, 5a732fbb5542992359bc3271, 5a8024ad5542992097ad2fde,
5ae142a4554299422ee9964a, 5a72d51555429991f9a20c5b4, 5a722a4b55429971e9dc931f, 5a7a9ca455429941d65f26f3,
5adfa5405542992d7e9f93ca, 5a7b8e3d55429927d897bfec, 5a7c6ac25542996dd594b925, 5abae9cd5542996cc5e49f04,
5ae18e37554299234fd0428f, 5a84d29d5542994c784dda60, 5ae44eeb5542995dadf2430f, 5adbe7b455429944faac23b0,
5abedd105542993fe9a41d63, 5a80a7df554299485f59867f, 5ab2f6b1554299545a2cfaea, 5ac29ddc554299657fa28fdc,
5a7222ce55429971e9dc92c7, 5ae221f15542994d89d5b366, 5a7f9cc25542995d8a8ddec2, 5abe42aa55429976d4830ac2,
5ae329e45542991a06ce993e, 5a882caa5542997e5c09a596, 5ac1a94455429944131be262, 5a762e0f5542992d0ec06052,
5a7918ec5542991481f19ef, 5a7e0bd25542997cc2c4750b, 5ab8af3c55429916710eb0ac, 5aba944655429994dbf019953,
5a82ef725542995ce29dcd0a, 5ab2a5fb554299545a2cf9ef, 5ab3d4ae5542992ade7c6ec5, 5ac25882554299636651998c,
5ae535f55542993aec5ec17c, 5ac55c915542993e66e8234f, 5adfcf7655429906c02daa49, 5a8a12555542992e4fca84f1,
5a8aff82c55429950cd6afc31, 5a8c564b554299240d9c2128, 5a89efb25542992e4fca8497, 5ab58009554299637185c5b2,
5ae69a455542996d980e7c48, 5a8f8dfb5542997ba9cb32bb, 5a811e1955429903bc27b931, 5a81f2955542990a1d231eee,
5abc428955429959677d6a67, 5ac263a255429921f1f2b38a3, 5ac5190d5542996feb3fe9f8, 5a82fbfc55429954d2e2ebe5,
5abce73b5542993a06baf9a2, 5adbf672554299438c868cf0, 5a75dd02554299109176e5aa, 5a8200d055429926c1cdade2,
5a8090105542996402f6a55c, 5adfda36554299025d62a35e, 5a7f9e0155429969796c1aee, 5a75f64554299042af8f757,
5a8a7bfb5542996c9b8d5eff, 5ae73fae5542991bbc9761c9, 5a77b0795542992a6e59df89, 5ac178655542994ab5c67d5a,
5ab5eab35542992aa134a3dd, 5ab667be55429954757d328a, 5a7a333f5542996a35c17130, 5ac262a055429951e9e6859a,
5a87ae9d5542994846c1cdc6, 5ac1985e55429964131be248, 5a848c215542992a431d1a4f, 5a89a79c5542993b751ca970,
5a8e16d355429917b4a5bd18, 5a7289755542992359bc30d9, 5a7d1dd055429909bec76960, 5ac152e755429964131be1bb,
5ae7d4f45542994846c5a659, 5a21559554299492dc91bc2, 5a8935e6554299669944a506, 5b831cb955429966c78a6b3f,
5a77aa565542992a6e59df6a, 5abff5e95542997d6429596a, 5ae07634554299603e418412, 5ab4eb2b55429942dd415fa2,
5abd512655429924427fcfb4, 5a7ad0195542992d025e66fd, 5a7cf9b455429907fabef07c, 5ae0fa52554299422ee99594,
5ae24d1a55429922decbdcca6, 5a7144df5542994082a3e72f, 5ac0279c5542996f0d89cb3f, 5a88a93c5542994846c1cead,
5adec5955542992fa25da83f, 5abbfd00554299114383a0d4, 5a7b9cac554299042af8f78f, 5ab9020d5542991b5579f0ca,
5a7c1c595542990527d55456, 5a7c583e5542996dd594b910, 5a8e72f05542990e94052b13, 5a85a1015542991dd0999e6f,
5adcb8205542994ed6169bd2, 5a8cef7a554299441c6b9f8a, 5a7fee435542994857a7685b, 5a7b4f2c55429931da12ca66,
5abeaf8a5542997ec76fd346, 5abbe67e5542993f40c73c05, 5a8f4e8955429918e830d1f1, 5ac1a0e15542994ab5c67dab,
5a7a9b4755429941d65f26ef, 5a87c1ac5542997e5c09a565, 5ab962ff554299131ca4231f, 5a7b79c95542997c3ec971b0,
5abe3ac35542993f32c2a0ac, 5a7639d55542992db9473748, 5a7a2ec05542990198eaf0bc, 5ac3d31a5542995ef918c249,
5abae3eb5542996cc5e49ee2, 5adff38b55429925eb1afb7d, 5ab7530b55429928e1fe3849, 5a88dcef55429938390d3fe3,
5ae0027b55429942ec259bda, 5a85ec815542994775f606af, 5ac172a15542994d76dcce2e, 5ac073eb5542996f0d89cbd8,
5ac5262755429924173fb60f, 5a8e72fe5542990e94052b14, 5a76133755429976ec32bcff, 5ae6b38c5542992ae0d16392,
5ab98fee554299131ca4237c, 5ac0e564554299294b219045, 5a72edeb5542992359bc31da, 5a7b663355429931da12ca87,
5a7cbe0f554299bec767ee, 5a845bdd5542996488c2e524, 5a8a28b55542996c9b8d5e23, 5ae5fb975542996de7b71aa8,
5aba9cff5542994dbf01997e, 5ae11f0b5542997b2ef7d0e0, 5abe16c655429976d4830a71, 5abbdd355542992ccd8e7fc6,
5abedbfa5542993fe9a41d5f, 5a792421554299148911fa09, 5a80c5f655429260e20a151, 5ab4136b5542996a3a969f18,
5adc375055429944faac246c, 5ac14d9d55429964131be1ab, 5abf23a655429477ec76fd3d7, 5ac1d4255429965cec5ea79,
5ae63c8f5542992663a4f27c, 5ae71816554299572ea546d1, 5ae4bdeb55429913cc2044ee, 5ae4a09e5542996836b02ced,
5ac2312755429964131be2c3, 5ae36d325542992e3233c3f8, 5a7d68045542995f4f40226d, 5aba88d555429901930fa811,
5a8e1e4b554299068b959e63, 5a7e6d325542991319bc94a7, 5ab96d865542996be20204df, 5ae4d2c255429960a22e01f6,
5a8053cf5542992097ad2fe0, 5a8db1b75542994ba4e3dd01, 5a8d40c95542994ba4e3dc3b, 5ae5af10554299546bf82f23,
5a8d48ff5542994ba4e3dc5a, 5ab5f694554299488d4d9a66, 5a8f99bc55429918e830d28d, 5add0ed35542990d50227dac,
5a8c38235542995e66a4755f, 5ab6ccf155429954757d3372, 5ae44fe755429945dadf24314, 5adcb67e5542994ed6169bca,
5abe833d5542993f32c2a140, 5a8b002155429950cd6afc3e, 5a76f3c65542994aec3b719a, 5ab5207c5542996a3a96a02b,
5a8a73dd5542996c9b8d5eee, 5a9063c955429933b8a2050f, 5a7b45c855429931da12ca4a, 5a8e8b6c5542990e94052b43,
5a7a57935542990783324f1d, 5abe225c5542991f661060ec, 5a72a6b65542994cef4bc3b7, 5a7f3625542995dae37ea06,
5a7cfdda55429907fabef095, 5a8994505542993b751ca950, 5ae308775542992decbdcdcd, 5ab72f32554299110f219ac3,
5a7b93e05542995eb53be961, 5a88710b554299206df2b26b, 5ab6259855429953192ad272, 5ac29ca6554299218029dac0,
5ac0ab335542992a796ded5d, 5ade469c5542992fa25da722, 5ab318a0554299233954ff07, 5ab1f75d554299340b525443,
5ade5664554299728e26c6d5, 5ae4a3b65542995ad6573dee, 5ae40e3955429970de88d8c5, 5ab9025855429934fafe6e47,
5a82100955429926c1cdae1e, 5ac5138c5542994611c8b36a, 5ab2eb7755429929539468b9, 5a73894554293667793f97



## CofCA Example Ids

5a866fee5542991e77181657,
5ac4bfd05542997ea680caab,
5ab381b155429969a97a816b,
5abaa25155429901930fa868,
5add82fc5542997545bbbd57,
5a79c7f95542994bb9457099,
5ae136f655429920d5234325,
5ae77a31554299540e5a55c7,
5a804fc45542992bc0c4a6f0,
5ae61be055429929b0807ace,
5ae7b03e5542993210983ef6,
5a866fee5542991e77181657,
5add28c85542992ae4cec4be,
5abba27f5542996606241708,
5a7336d05542991f9a20c68d,
5ae6038155429929b0807a55,
5a76cb6e5542994aec3b717a,
5ae0fa8b5542997b2ef7d0c6,
5ac491eb5542996feb3fe8d2,
5ac219df5542992f1f2b37fc,
5a8e171b554299068b959e5a,
5adce28f5542990d50227d52,
5a80043055429969c1ba0,
5a8f7de3554299458435d657,
5adc318c5542996e685252d5,
5abaf9df5542996cc5e49f45,
5a83168855429966c78a6b2e,
5abbdc135542993f40c73bf6,
5ab3dad4554299753aec59cb,
5ac143535542991316484aac,
5abd04f15542996e802b467e,
5ae061d55542996303e41840e,
5a7a358f5542990783324ec1,
5a8ee0a35542990e94052ba0,
5a7bb3d9554299294a54aaa0,
5add992c5542997545bbbd83,
5a8b58b955429950cd6afcc2,
5a8d0006554299441c6b9fa8,
5a81ff1d554299676cceb1c3,
5abbbd0f55429931dba144d5,
5ab5141a5542991779162d70,
5ab698885542995eadef002a,
5ae377155542991a06ce99c7,
5ac168865542994ab5c67d14,
5ae27edc5542992decbdcd2d,
5ae3b4d05542992f92d82349,
5a88b7735542993e715ac079,
5ac538ef5542994611c8b437,
5ae31a9c55429928c42395ef,
5ade52e85542997c77adedfa,
5a7fa53c5542995d8a8ddedc,
5ab55435554299488d4d9939,
5ab3dde2554299753aec59d6,
5ae755665542997b22f6a6e9,
5a8099025542996402f6a588,
5ae13f525542997b2ef7d169,
5a760f6855429976cec32bcf9,
5a90c2b35542995651fb51df,
5ac168865542994ab5c67d14,
5ac219df5542992f1f2b37fc,
5a879ab05542996e4f30887e,
5ae5365d5542992663a4f16d,
5ab9121555429919ba4e238a,
5ae5365d5542992663a4f16d,
5ac31c9d554299741d48a203,
5a7cc5ae55429909bec767fc,
5a77aff55542992a6e59df86,
5ae77a31554299540e5a55c7,
5ae136f655429920d5234325,
5ae1f61a5542994d89d5b2e1,
5a7336d05542991f9a20c68d,
5ae2aba15542996483e64a32,
5a736bfa5542991f29ee2e03,
5ab6e856554299710c8d1fac,
5ac4fa8c55429924173fb536,

5a7db2f75542990b8f503a34,
5ac4c61a5542996feb3fe93c,
5ac2a912554299218029dae8,
5ac39a1c554299657fa290f9,
5add117e5542990d50227db2,
5ab58ae15542992aa134a357,
5a80b4635542992bc0c4a7bd,
5ae0d91e55429924de1b7198,
5ac2ffa9554299218029dbb2,
5a8ae6c055429950cd6afbce,
5a8eacc75542995085b37473,
5a888a8a5542997e5c09a603,
5abffc58554299012d1db552,
5ab6ad2855429901930ad35e,
5ac3b0f15542995ef918c1fc,
5ab3dde2554299753aec59d6,
5a7524ca55429929fddd850a,
5a7336d05542991f9a20c68d,
5a7fe9975542994857a76847,
5a8a84775542996c9b8d5f19,
5a79e0445542994f819ef0e7,
5a8f7de3554299458435d657,
5ac39f2a554299391541382d,
5ab74412554299110f219ae8,
5a78cdf7554299029c4b5e9f,
5adcf28c5542994ed6169c30,
5adc134b5542994650320c5c,
5add7d055542990dbb2f7e61,
5a8dcd8e55429941ae14e060,
5ac31c9d554299741d48a203,
5a72b2695542991f9a20c56f,
5ae56d31554299546bf82ed7,
5a7f38ae5542992e7d278c99,
5ae3bdfa5542990afbd1e1c0,
5abb1f745542996cc5e49fb5,
5adc1309554299438c868d3b,
5ae22d035542996483e64925,
5add82fc5542997545bbbd57,
5ae755665542997b22f6a6e9,
5a7ccec9554299452d57ba72,
5a7f98e655429969796c1ad8,
5ae614055542996de7b71b2a,
5ac1944c5542996f0d89cc90,
5ab979da554299131ca4233a,
5ab39701554299233954ff5e,
5adfd35b55429906c02daa54,
5ab520565542996a3a96a02a,
5ab67b8f55429954757d32f0,
5a7607d7554299109176e61a,
5abaef34554299660624169c,
5a904e725542995651fb5118,
5ac3ad225542995ef918c1da,
5ae316f355429928c42395e3,
5a74248855429929fddd83e5,
5ac1f7f355429964131be2ae,
5a7f38ae5542992e7d278c99,
5a78ed46554299148911f9a6,
5a88b7735542993e715ac079,
5ab67b8f55429954757d32f0,
5ae5365d5542992663a4f16d,
5a736bfa5542991f29ee2e03,
5a8dfbeb5542995085b3736e,
5add28065542990d50227e08,
5adf2b325542993a75d2640b,
5a8a84775542996c9b8d5f19,
5ae31a9c55429928c42395ef,
5ac2adf3554299657fa2900f,
5ae525835542990ba0bbb1cd,
5add28c85542992ae4cec4be,
5a8eacc75542995085b37473,
5abba27f5542996606241708,
5a8f0e065542997ba9cb319c,
5aba0e675542994dbf0198a0,
5a8beddd5542995d1e6f1468,

5a8ee0a35542990e94052ba0,
5abbbd0f55429931dba144d5,
5ae3345f55429928c4239682,
5a73332b5542992359bc3287,
5ab93287554299753720f78f,
5ae3bdfa5542990afbd1e1c0,
5ab93287554299753720f78f,
5ae64cab5542991bbc9760be,
5ae7b03e5542993210983ef6,
5ae64cab5542991bbc9760be,
5ab9253c554299131ca4227f,
5abeed7e5542993fe9a41da0,
5a8bab4e5542992a40d9c207c,
5aba6b2d5542990130fa7a9,
5ac3ad225542995ef918c1da,
5ab381b155429969a97a816b,
5ade025e5542997dc790711e,
5a8b560855429950cd6afcba,
5a72b2695542991f9a20c56f,
5abd7ca05542993062266cab,
5ae0d26455429945ae959473,
5adc1309554299438c868d3b,
5a72b1c25542992359bc3172,
5a904e725542995651fb5118,
5ade8f5e55429975fa854f11,
5a7e7bf455429949594199d6,
5a90c58255429916514e756c,
5ab344af554299753aec5969,
5ae377155542991a06ce99c7,
5ae5569255429908b63265e4,
5ab59b045542997d4ad1f190,
5ae255db55429932decbdccc1,
5ab5c9c5554299494045f065,
5ab561d85542992aa134a2fc,
5adce28f5542990d50227d52,
5adfd35b55429906c02daa54,
5a7fa53c5542995d8a8ddedc,
5a80d30655429938b61421fe,
5a79e0445542994f819ef0e7,
5a7bb3d9554299294a54aaa0,
5ae2e27155429928c423952a,
5a77bd595542995d83181291,
5a823ae45542990a1d231f6d,
5a7cedca55429909bec7689c,
5ab345db55429969a97a8122,
5add992c5542997545bbbd83,
5a8514545542992a431d1ad2,
5a74fbe55542996c70cfae63,
5ae13f525542997b2ef7d169,
5a85603a5542997b5ce3fff1,
5ae3d8dc5542992f92d8239c,
5a879c8e5542994846c1cdb3,
5ae11a6755429901ffe4ad8d,
5abfbb455542997ec76fd440,
5ac39a1c554299657fa290f9,
5a84322b5542996488c2e50d,
5ae655c855429908198fa599,
5a8454e85542992ef85e23be,
5a77aff55542992a6e59df86,
5a7a0d455542990783324e13,
5abfbb455542997ec76fd440,
5a8a35625542992d82986efd,
5ae64cbf5542992ae0d162c1,
5ae755665542997b22f6a6e9,
5ae377a35542994393b9e6db,
5adf5ebd5542995ec70e8fd8,
5ab5a2f85542997d4ad1f197,
5a7738dc5542993735360021f,
5a8bdef85542997f31a41dea,
5a8cdc5255429941ae14d21,
5abd7ca05542993062266cab,
5a8a2d805542996c9b8d5e2e,
5ae3345f55429928c4239682,
5abd90545542996e802b47d7,

5ae525835542990ba0bbb1cd,
5a89372855429951533612e6,
5a7bb3d9554299294a54aaa0,
5ae655c855429908198fa599,
5ab979da554299131ca4233a,
5add7d055542990dbb2f7e61,
5ae3b4d05542992f92d82349,
5ab865be5542990e739ec8e5,
5a77153355429937353601c8,
5a72a00d5542991f9a20c53c,
5a8ee0a35542990e94052ba0,
5ae4b3da55429913cc2044d6,
5abae52a5542996cc5e49eea,
5abc145b55429966658360041f,
5a7a06935542990198eaf050,
5a77bd595542995d83181291,
5ac17f4f5542994ab5c67d70,
5adce28f5542990d50227d52,
5a89372855429951533612e6,
5ac07a585542996f0d89cbf0,
5a8f0e065542997ba9cb319c,
5ac219df5542992f1f2b37fc,
5abe3f9455429976d4830aaa,
5a7a02235542996c55b2dcd3,
5ab865be5542990e739ec8e5,
5a8efd3c55429918e830d179,
5a8a35625542992d82986efd,
5ab93287554299753720f78f,
5a7f3d325542992e7d278cb5,
5ab6e856554299710c8d1fac,
5ac061ab554299294b218fac,
5ae3d8dc5542992f92d8239c,
5a904e725542995651fb5118,
5ab39701554299233954ff5e,
5a84322b5542996488c2e50d,
5a72b2695542991f9a20c56f,
5ae4c2145542995dadf243e7,
5a8355f9554299123d8c20f3,
5abee5e25542994516f45473,
5a78ed46554299148911f9a6,
5ab520565542996a3a96a02a,
5a88fea05542997e5c09a6e9,
5ab5e6d65542997d4ad1f232,
5adfd35b55429906c02daa54,
5ab55435554299488d4d9939,
5a7d1f605542995ed0d165fb,
5ac17f4f5542994ab5c67d70,
5ae0fa8b5542997b2ef7d0c6,
5a870d0255429960ec39b710,
5ab9116f55429991b5579f0db,
5a88377c5542997e5c09a5a7,
5abbc70d5542992ccd8e7f9b,
5a7738dc554299373536021f,
5a821ffa5542990a1d231f5c,
5a8514545542992a431d1ad2,
5ab39701554299233954ff5e,
5a8461d55542990548d0b29b,
5a7f9ee855429969796c1af3,
5a8f8f345542997ba9cb32c2,
5ac31c9d554299741d48a203,
5adc134b5542994650320c5c,
5a8454e85542992ef85e23be,
5ac4fa8c55429924173fb536,
5a8a4bdc55429930ff3c0d8c,
5abd7cb855429924427fd00a,
5a7a52745542996c55b2dd4f,
5ae614055542996de7b71b2a,
5ae664955542992ae0d1631b,
5ac1a5cd5542994d76dcce94,
5ae546e85542992663a4f1b5,
5a7a02235542996c55b2dcd3,

5a7e39515542995ed0d166da

**MuSiQue Example Ids**

2hop__376129_44537, 2hop__764465_126539, 3hop1__434518_136629_55288, 2hop__353084_36340, 2hop__344450_160798, 2hop__637856_351187, 2hop__760990_44191, 3hop1__162325_11248_3752, 2hop__326799_278127, 2hop__239927_62031, 2hop__153813_69936, 3hop1__213491_782843_75255, 2hop__2846_2741, 2hop__3880_909, 2hop__347735_36735, 2hop__144393_87372, 4hop1__709382_146811_31223_45305, 2hop__143434_20122, 2hop__21457_74218, 3hop1__129597_517267_451901, 2hop__469317_776926, 2hop__27032_5400, 3hop2__83954_32417_24628, 3hop2__14790_57411_86234, 2hop__78490_49700, 3hop1__228008_354329_5303, 2hop__631861_160851, 3hop1__662283_507729_351187, 2hop__482727_20661, 3hop1__858308_102146_84004, 2hop__565717_77346, 3hop1__470555_668347_492654, 2hop__25478_65517, 2hop__129389_31248, 2hop__527889_5365, 2hop__20857_20779, 2hop__770_919, 2hop__375649_80178, 3hop1__332614_131794_17114, 2hop__144295_211364, 2hop__108160_159045, 2hop__46545_88521, 2hop__518906_44191, 2hop__733628_131886, 4hop1__28235_74795_84660_15312, 2hop__104341_92821, 2hop__445544_127008, 2hop__46766_79233, 2hop__342213_185893, 2hop__528837_126102, 2hop__497897_541630, 3hop1__48619_26424_581618, 2hop__87287_83906, 4hop1__411538_805015_475503_32631, 2hop__658198_72962, 2hop__42307_120207, 2hop__30878_555599, 3hop1__8373_87072_45358, 3hop2__337255_48727_83343, 2hop__251450_8796, 3hop1__161080_639509_644660, 2hop__558231_52667, 2hop__424189_49441, 3hop1__821692_74047_756423, 3hop1__531731_79705, 3hop1__257981_259472_611044, 2hop__370765_14904, 2hop__446352_14183, 2hop__81087_13292, 2hop__684971_333904, 2hop__234176_69926, 2hop__858097_121880, 4hop2__724536_444580_75897_631997, 2hop__492509_70585, 4hop1__405751_4520_65397_49736, 2hop__128610_126060, 3hop1__325154_786384_42990, 2hop__34130_56335, 2hop__145997_63766, 2hop__146446_690423, 2hop__225632_11125, 2hop__856457_495, 2hop__129234_330515, 2hop__15674_42467, 3hop1__161946_84298_53741, 2hop__48959_83539, 2hop__64650_20556, 3hop1__316518_395352_131877, 2hop__136618_92216, 2hop__199336_185893, 2hop__930_57555, 3hop1__31942_48661_15069, 2hop__35105_160978, 2hop__128804_351187, 2hop__153004_86587, 2hop__715365_565667, 2hop__401484_135138, 2hop__52622_67783, 2hop__713501_58946, 2hop__300786_39199, 2hop__5430_5348, 3hop2__29467_132027_73594, 3hop1__225298_755188_480696, 2hop__367037_80178, 2hop__343473_53204, 2hop__848923_66214, 3hop1__369072_287321_161879, 2hop__250315_64214, 3hop1__104311_833580_61459, 2hop__1835_322987, 3hop1__836616_291186_4303, 2hop__531924_1094, 2hop__131831_84128, 2hop__328708_90697, 2hop__704691_82816, 2hop__80353_3001, 2hop__196785_61424, 2hop__130964_47336, 3hop1__761109_548045_159613, 3hop1__4525_52205_55099, 3hop1__58522_787757_69397, 2hop__58284_37793, 2hop__487591_7672, 2hop__250913_58115, 2hop__131095_85298, 2hop__144937_8600, 2hop__625639_25582_21116, 3hop2__30023_63595_53125, 2hop__584872_88978, 2hop__116643_351162, 2hop__826203_62031, 2hop__85036_909, 2hop__62996_299942, 2hop__236731_229413, 2hop__15169_87091, 2hop__143791_75878, 2hop__658198_90536, 2hop__70321_15755, 2hop__131105_68117, 2hop__143162_438686, 2hop__20771_65517, 2hop__65149_46180, 2hop__238983_403313_61770, 2hop__28291_709757, 2hop__391909_3430, 3hop1__266733_291186_50964, 2hop__205685_160137, 2hop__343141_702969, 3hop1__383692_434040_59381, 2hop__240975_736878, 2hop__507864_368521, 3hop1__723003_593059_76293, 2hop__109234_62766, 4hop1__16401_4520_65397_52251, 2hop__140591_256194, 2hop__104757_74309, 2hop__194976_55566, 2hop__361127_140822, 3hop1__108774_104782_14771, 4hop3__393686_620110_61746_261712, 2hop__324178_83854, 3hop1__849536_301867_127418, 2hop__24408_541630, 2hop__54755_729624, 2hop__693650_61232, 3hop1__89787_49283_632017, 4hop1__104663_221169_833580_61459, 2hop__664573_36741, 3hop1__702271_823374_26254, 2hop__129892_62851, 3hop1__659125_39490_23352, 2hop__222162_386543, 2hop__446009_412262, 2hop__781841_77980, 3hop1__706183_20196_10585, 2hop__809948_162428, 3hop1__458602_681261_369731, 2hop__529082_114112, 3hop1__388966_508834_145463, 2hop__582169_370960, 2hop__225632_52135, 2hop__302491_81463, 2hop__136889_52356, 2hop__81363_42667, 2hop__599980_544161_92922, 2hop__504710_513189, 2hop__145939_11443, 3hop1__320353_4018, 2hop__27033_85063, 2hop__145110_861627, 2hop__149891_44359, 2hop__376266_37939, 3hop2__10879_37094_161133, 3hop2__159915_8509_19700, 4hop1__15118_31258_43153_32993, 3hop1__522518_132413_16066, 2hop__129782_517267, 3hop1__252998_715836_26008, 4hop1__205937_144938_83779_44678, 2hop__131318_47465, 2hop__338405_68172, 4hop3__3153_3356_11988_24628, 2hop__106465_54210, 2hop__397761_404718, 4hop1__632232_164954_6975_6891, 2hop__121872_708662, 2hop__73501_31113, 2hop__378511_191233, 3hop1__85045_96305_25007, 3hop1__755950_592709_78102, 2hop__811421_377891, 3hop2__63595_391767_53125, 2hop__131380_84859, 3hop1__158678_48408_37793, 3hop1__7312_830682_68600, 2hop__207212_21032, 3hop1__10725_695397_74345, 2hop__445228_774871, 4hop1__603090_818753_783943_26110, 2hop__177131_646483, 3hop1__801682_192919_16121, 2hop__243908_500443, 3hop2__89818_157704_4107, 2hop__160546_26427, 2hop__128772_745471, 2hop__62588_20779, 2hop__661636_82027, 2hop__105388_89066, 2hop__368185_131944, 3hop1__153577_411195_8682, 2hop__327451_90697, 2hop__647590_134798, 3hop2__30796_804098_24137, 2hop__146227_42328, 2hop__152881_620955, 2hop__11693_42892, 2hop__753498_7606, 2hop__2795_2741, 3hop1__373317_533132_1660, 2hop__229374_333904, 3hop1__370820_301867_127418, 3hop1__713250_4016_83854, 2hop__130414_68117, 4hop1__7312_84360_334118_41330, 2hop__65149_68376, 2hop__182310_565529, 3hop1__136299_84467_89676, 2hop__454055_86874, 2hop__604878_40786, 2hop__307569_51671, 2hop__854082_159115, 2hop__198557_55566, 3hop1__352446_506157_44678, 2hop__468848_44537, 2hop__207571_126101, 4hop2__53235_18485_57802_311656, 2hop__451164_140822, 3hop1__37692_84298_53741, 3hop1__672119_196807_760519, 3hop2__131210_661360_54023, 2hop__8531_24846, 3hop2__77886_64137_69951, 2hop__730762_8600, 2hop__350323_45731, 2hop__131117_53519, 3hop1__157534_275705_81669, 2hop__185628_677577, 2hop__77119_20732, 2hop__67755_82010, 3hop1__790278_593059_76293, 3hop2__162189_611045_73761, 2hop__568848_50788, 2hop__45625_61952, 2hop__146207_30651, 2hop__57439_78714, 2hop__3756_52135, 3hop1__501828_348668_856982, 3hop1__106423_35178_686699, 2hop__103203_23140, 3hop1__77985_66386_16350, 2hop__664921_579740, 2hop__106125_20644, 2hop__400998_61424, 3hop1__35884_161545_16532, 2hop__584521_755188, 2hop__80508_400874, 2hop__664137_58115, 2hop__453207_80674, 3hop1__29335_30907_24600, 2hop__144364_68900, 2hop__226817_482901, 4hop3__39198_75897_8509_19700, 2hop__713863_64008, 2hop__71269_36735, 2hop__504228_64689, 2hop__604878_18657, 2hop__81372_303417, 3hop1__674688_707133_72062, 2hop__157766_18657

