# OpenReview forum: "Synthetic Data Generation and Multi-Step Reinforcement Learning for Reasoning and Tool Use"
_colmweb.org/COLM/2025/Conference — COLM 2025_

### Official Review · Reviewer_SNTP · 2025-05-05

**Rating:** 7
**Confidence:** 4
**Ethics Flag:** 1

**Summary:**

This paper presents SWiRL, a multi-step optimization technique. SWiRL consists of two stages. The first stage involves generating multi-step synthetic data. The second stage is the learning stage: learning from this data in a step-wise reinforcement learning style.

The first stage is the generation of multi-step synthetic trajectories (s1,a1,…,sk,ak): a sequence of states and actions where the first state is the prompt to a language model, subsequent states build context from the previous state, action, and environment response—actions are the model's responses to the current state—and the last action ak is declared as the (final) answer to the original prompt. The model has the option at each step to either invoke a tool (e.g. a calculator) or give a final response, and it usually produces chains of thought. When the model generates a tool use call, the system parses it, executes it in the environment, and presents the result to the model in the next step. In training, based on the preceding context, the base model is optimized at each step to predict the next step or the final response.

The paper presents four filtering approaches: none, outcome-based, process-based (model-judged steps), and combined process and outcome-based. There are several multi-hop question-answering and mathematical reasoning datasets used in experiments.
While some recent approaches filter synthetic data based on correct final answers, the authors argue that SWiRL benefits from training on the data that is not filtered for the correctness of final answers. This boosts the accuracy of LLMs on average by 15 percent. Moreover, the authors suggest that filtering for correctness usually impairs performance.

**Reasons To Accept:**

This paper has an interesting topic that should definitely appeal to computational linguistics: boosting the performance of a language model on reasoning tasks without using human annotated labeled data but synthetic one.
The approach is intriguing because it's different from how synthetic data has been used in recent methods (the authors argue that filtering for correct final answers harms the performance).
It is a well written paper.

**Reasons To Reject:**

While I believe that this paper should not be rejected, to me it seems somewhat necessary to have a more detailed analysis of the obtained results. Maybe trying it out on a few more datasets could help the authors to understand why the unfiltered data seems to work better than the filtered one.

---

> ### Author Response · Authors · 2025-06-03
>
> Thank you for your careful review of our paper and for your helpful and insightful comments! We also appreciate your kind words regarding the writing quality and the novelty of our approach to leveraging synthetic data.
>
> Regarding intuition for why unfiltered data outperformed correctness-filtered data, we share our thoughts and new results below:
>
> * As discussed in the paper, we found that filtering for correctness harmed performance (compared to not filtering), but that filtering for "reasonableness" of each step improved results. It is worth noting that SWiRL training produces performance gains regardless of filtering strategy, but the magnitude of those gains varies based on filtering strategy.
> * This is a somewhat surprising result, but our intuition is that filtering for correctness effectively means that we are only training on examples which the model can already answer correctly, and that the model learns more when it is also exposed to examples which it has not yet mastered.
>
> Thanks again for your helpful suggestions and feedback, which we will use to improve our paper and its presentation!

---

> > ### Comment · Reviewer_SNTP · 2025-06-06
> >
> > Thanks. Given that it is not evident how one can define what is the “reasonableness” filtering in general (and, in particular, in the cases where the goal was not achieved), whereas it is slightly easier to define what is correct (achieving the goal) vs incorrect (not achieving the goal), it would be desirable to have this approach tested on more kinds of data, at least as part of the future work.

---

### Official Review · Reviewer_SiLC · 2025-05-09

**Rating:** 7
**Confidence:** 3
**Ethics Flag:** 1

**Summary:**

### First, I want to mention that I am not familar with both RL and Tool use in LLMs. Sorry about that. If I make some errors, just let me know. Thank you for AC and authors.

The paper introduces Step-Wise Reinforcement Learning (SWiRL), a novel approach designed to enhance large language models in performing complex, multi-step reasoning tasks that involve tool usage.

**Reasons To Accept:**

- The decomposition of trajectories into sub-trajectories for step-wise optimization is a novel approach that allows for granular learning and feedback.

- By relying on synthetic data and model-based evaluations, SWiRL reduces the need for costly human annotations, making it a scalable solution.

**Reasons To Reject:**

- The paper does not provide a comprehensive comparison between SWiRL and other contemporary approaches to multi-step reasoning and tool use in language models. Including such comparisons would offer clearer insights into SWiRL's relative advantages and limitations.

- SWiRL relies heavily on synthetic data generated by language models and filtered through model-based evaluations. This approach may introduce biases or inaccuracies inherent in the models themselves, potentially leading to suboptimal learning outcomes. The absence of human oversight in data generation and filtering raises concerns about the reliability of the training data.

- I guess I miss something, but I didn't find the model used in generated training data. What model did you use, and is there the results to show whether this method can be used in self-improve?

---

> ### Author Response · Authors · 2025-06-03
>
> Thank you for carefully reading our paper and for your thoughtful comments! Regardless of your background in RL and tool use, we appreciate your feedback and will strive to address it to your satisfaction.
>
> We also appreciate your kind words regarding our novel approach to decomposing multi-step trajectories into a single-step RL optimization problem, and the scalability of our synthetic data generation approach.
>
> * Although prior approaches have been proposed for multi-step reasoning (e.g. DQO, OREO, and PRIME), we are not aware of any approaches that specifically tackle the multi-step tool use problem (or at least none published prior to the submission of our article to COLM). We therefore felt that SFT represented the most directly comparable approach to leveraging the synthetic trajectories we generated. We also compared against the base model in order to measure the impact of such post-training (see Table 1 and newly added Appendix A) and compared against the reward model (newly added Figure 8). See https://imgur.com/a/O4tt0bq and https://imgur.com/a/Wxymhkz.
> * The synthetic data is generated by the base model itself (the model being trained). While it is true that the synthetic trajectories in question may contain hallucinations or may be sub-optimal in quality, we do filter these trajectories using a PRM (which checks that each step represented a reasonable action given all actions taken so far), improving final performance (though we still observe improvements, albeit weaker ones, when no filtering is performed).
> * Interestingly, we ran experiments in which we trained on trajectories generated by a stronger model, rather than the weaker base model, and found that this actually degraded performance, which is likely due to the "off-policy"ness of training in this manner (i.e. it is more helpful for the model to operate and receive feedback in the context of its own current capabilities / distribution, rather than learning about what would happen if other models had taken actions in the past). In other words, there is significant value in having the contextual data from prior states be drawn from (approximately) the same model as that which is being trained. See https://imgur.com/a/impact-of-synthetic-data-source-on-performance-m5hkqP5.
>
> Thanks again for your thoughtful review and feedback which we have used to improve the clarity and presentation of our work. Please let us know if you have any other questions or suggestions!

---

> > ### Comment · Reviewer_SiLC · 2025-06-03
> > **Thanks authors' response**
> >
> > Thank you for your response. The finding is quite interesting and also expolred in "Small Models Struggle to Learn from Strong Reasoners". I have updated the confidence and score.

---

### Official Review · Reviewer_4922 · 2025-05-13

**Rating:** 8
**Confidence:** 4
**Ethics Flag:** 1

**Summary:**

The authors propose a simple and effective RL method that enables LLMs to make decisions during CoT reasoning and tool use, aiming to enhance the multi-step reasoning abilities of LLM agents. Specifically, the synthetic rollouts are generated from a base LLM interacting with tools. Then the rollout receives an outcome reward based on the final result and a process reward generated by a stronger LLM. These rewards are then used to update the model. Experiments demonstrate the effectiveness of the proposed approach.

**Questions To Authors:**

1. Regarding the generalization across tasks, has the author analyzed the reason for this OOD improvement?

**Reasons To Accept:**

1. The proposed method effectively demonstrates the usefulness of PRM in LLM agents’ tool use. The idea of learning from imperfect but informative trajectories is also insightful for future work.

2. The method is evaluated on multiple tasks and shows good OOD generalization.

3. The performance improvement over the base model and SFT is substantial.

**Reasons To Reject:**

1. A potential limitation is that the process reward relies entirely on a single strong model (Gemini 1.5 Pro Thinking) for scoring. Although this model is strong enough to support the current experiments, it narrows the conclusion. In broader task settings, particularly those involving harder problems or less generalizable reward models, the accuracy of process evaluation might not be reliable, potentially compromising RL optimization. As a result, the stability of the SWiRL method still largely depends on the quality and selection of the reward model, and the conclusion may not be stable for other tasks or reward models.

2. In addition, the current tool selection is still relatively over-simplistic, which may make the reward estimation easier. For example, if multiple search tools were involved, evaluating the process reward would become more challenging as it needs to check which searching tools are appropriate for this certainty question.

---

> ### Author Response · Authors · 2025-06-03
>
> Thank you for your careful review of our paper and your thoughtful comments! We are glad that you found it interesting that we could learn from imperfect trajectories and that you felt our results were strong.
>
> 1) **Reliance on Strong Reward Model:** Although we rely on a stronger reward model, our strong generalization results across new tasks and tools at test time (where we do not have access to any reward model) indicate that SWiRL is indeed enabling the model to generate better reasoning steps and tool calls. To further validate this observation:
>     * We ran a new set of experiments to measure the capabilities of the current teacher model on the target tasks, as shown in the newly added Figure 8. We found that SWiRL training can enable the base Gemma 2 model to exceed the performance of the teacher model Gemini 1.5 Pro. This suggests that performance with SWiRL is not capped by the capability of the teacher model, and that this approach could potentially enable bootstrapping / model self-improvement, given a sufficiently strong starting point. See https://imgur.com/a/Wxymhkz.
>     * For what it’s worth, the reward model used (Gemini 1.5 Pro) is far from state of the art, but this approach is already effective for reasonably challenging and varied benchmarks (HotPotQA, GSM8k, etc.). Note that Gemini 2.5 Pro achieves 1446 on LMSys Text Arena compared to 1329 for Gemini 1.5 Pro.
>
> 2) **Overly Simplistic Tool Selection:**
> We agree that it would be interesting to explore more complex and varied tools. It is even possible that such settings would yield greater benefit from SWiRL than simpler tool use settings, as there may be more headroom for the model to learn how to effectively invoke such tools.
>
>     * In the newly added Figure 6, we show that the base models (both the teacher and student) already benefit from the proposed SWiRL inference (i.e. prompting the model to perform multi-step tool use). SWiRL RL training further enhances these capabilities. The gains could potentially grow if the tool was more difficult to use and therefore benefited more from training. See https://imgur.com/a/U0f9GRh.
>     * The two types of tasks on which we evaluated were quite different from each other, e.g. answering mathematical reasoning questions with access to a calculator vs. answering multi-hop questions with access to a search engine, and yet we still observed generalization across these different domains and tools. In addition to Table 2, in which we reported strong generalization results, we also ran new experiments to explore the impact of scaling out-of-distribution dataset size. Interestingly, as we increase the number of synthetic HotPotQA trajectories (multi-hop question-answering with search tool use) on which we train our policy, performance on GSM8K (multi-step math reasoning with calculator tool use) continuously improves at all dataset sizes explored. See an updated version of Figure 5: https://imgur.com/a/nBrWSrI.
>
> 3) **Mechanism Driving Generalization Across Tasks:** To provide insight into the mechanism by which SWiRL achieves OOD improvements:
>
>     * In the final subsection of "Results and Discussions", we looked into how SWiRL achieved these generalization results. In Table 3, we examined the average PRM scores before and after SWiRL optimization. We observe that both for in-distribution and out-of-distribution tasks, the SWiRL model generates trajectories with higher average process labels, suggesting that the performance gains and OOD generalization are driven by improvements in reasoning and capability to break down complex tasks into manageable pieces. See https://imgur.com/a/v6K0uQV.
>
> Thanks again for your helpful comments and questions! We have incorporated your feedback to substantially improve the paper and its presentation. Please let us know if you have any other questions!

---

> > ### Comment · Reviewer_4922 · 2025-06-07
> >
> > Thanks for the authors’ response. The experiments are impressive and have addressed my concerns, and I hope the results can be incorporated into a future version of the paper. I have updated my score accordingly.

---

### Official Review · Reviewer_WJVa · 2025-05-21

**Rating:** 6
**Confidence:** 4
**Ethics Flag:** 1

**Summary:**

* This paper proposes to apply multi-step reinforcement learning to improve LLMs' reasoning and tool use. While multi-step RL has already been explored in LLMs, the contribution of this paper should be considered as the introduction of a teacher LLM as a process reward model (PRM) to provide reward in the tool use area.

* The writing and clarity of this paper can be further improved. Some key contributions and novelties in the paper are not easy to capture.

* The experiment results are generally good, demonstrating the effectiveness of the proposed method in Gemma/ Gemini. One concern should be the limited number of teacher/ student models used in the experiment, which calls into question the generalization and robustness of the proposed method.

**Reasons To Accept:**

* This paper explores the effect of multi-step RL in tool use, which is an important area that should benefit from RL training intuitively.

* The experiment results are good. The authors demonstrate that their method improves the base model's performance significantly and outperforms vanilla SFT by a considerable margin. Additionally, some insights they provide about the selection of rejection sampling and reward methods can also benefit future works in that area.

**Reasons To Reject:**

* This work appears somewhat incremental and engineering-oriented. The authors integrate several existing techniques—multi-step reinforcement learning, data synthesis with rejection sampling, and the use of LLMs as process reward models—and apply them to the task of tool use.

* The experimental evaluation is limited to a single student-teacher model pair, which weakens the generalizability of the conclusions and findings.

---

> ### Author Response · Authors · 2025-06-03
>
> Thank you for carefully reviewing our paper and for your insightful comments! We appreciate your kind words regarding the value of this line of research.
>
> 1) **Novelty**: To address this concern, we have improved the writing and have added new figures and ablation studies to better highlight the novelty and effectiveness of our approach.
>     * In addition to incorporating model-based rewards during RL optimization, the manner in which we generate and filter the synthetic trajectories is also novel. We've added a new Figure 3 to more clearly demonstrate how this is performed. See https://imgur.com/a/ewXVb1X.
>     * We would also argue that the way in which we decomposed multi-step RL into single-step RL optimizations is both novel and effective. Not only does this formulation allow us to perform better per-step reward attribution and generalize across domains and tools, but it also enables offline execution of arbitrarily complex, lengthy, or unstable tools.
>     * Finally, as you noted, the tool use aspect is of growing interest and importance to the community, but remains relatively underexplored. Prior work on multi-step RL, such as DQO, OREO, and PRIME, did not focus on enhancing a model's ability to use tools or interact with an external environment. In contrast, we ran a new ablation study in which we measured the performance of the base model and the reward model with and without access to tools. As shown in the newly added Figure 6, SWiRL training significantly improves the model’s ability to effectively leverage external tools. See https://imgur.com/a/U0f9GRh.
>
> 2) **Generalizability**: To address concerns regarding generalizability of our results, we ran a number of new ablation studies, and surface relevant findings from our original appendix.
>     * In the original Appendix B, we varied the student model (Gemma 2 2b, Gemma 2 7b, Gemma 2 27b) to explore the effect of model size on the effectiveness of SWiRL. We found that SWiRL robustly improves performance for the 27b model across both in-domain (HotPotQA) and out-of-domain datasets (MuSiQue, CofCA, and BeerQA). However, while the in-domain improvements hold for smaller models, the out-of-domain performance is mixed, suggesting that the effectiveness of SWiRL grows with parameter count and that it may therefore be even more useful for training future models. See https://imgur.com/a/Jr0e23A.
>     * In an updated version of Figure 7, we explored the impact of scaling out-of-distribution dataset size on the performance of SWiRL. Interestingly, as we increase the number of synthetic HotPotQA trajectories (multi-hop question-answering with search tool use) on which we train our policy, performance on GSM8K (multi-step math reasoning with calculator tool use) continuously improves at all dataset sizes explored. See https://imgur.com/a/nBrWSrI.
>     * In a newly added Figure 8, we ran experiments in which we directly compared the performance of the student and teacher models, and found that SWiRL enables the student model to outperform the teacher in many cases (e.g. 3 out of 5 of the datasets). These results suggest that SWiRL is not merely distilling the stronger reward model, but rather enables generalization to tasks and tools beyond what the base model has been trained on, and even enables it to outperform the reward model in many cases. See https://imgur.com/a/Wxymhkz.
>
> Thanks again for taking the time and care to review our paper and for sharing these valuable insights! We have carefully considered your feedback and have used it to substantially improve the work and its presentation.

---

> > ### Comment · Reviewer_WJVa · 2025-06-03
> > **Thanks authors' response**
> >
> > Thanks to the authors' response to my review. It addresses my concerns and I keep my positive attitude toward this paper.

---

### Decision · Program_Chairs · 2025-07-08

**Decision:**

Accept

**Comment:**

This paper presents SWiRL (Step-Wise Reinforcement Learning), a method for improving multi-step reasoning and tool use in large language models through synthetic data generation and reinforcement learning optimization. The paper demonstrates clear technical contributions in the step-wise breakdown of multi-step trajectories into single-step RL optimizations that enables granular learning and better reward attribution, while reducing reliance on human annotations through synthetic data generation. The empirical results show substantial improvements across multiple benchmarks, including GSM8k and HotPotQA, with notable cross-task generalization. This confirms the potential of the method for reasoning tasks. While the tool environments are relatively simple, they are sufficient to demonstrate that the approach is workable.